# Resveratrol Contrasts IL-6 Pro-Growth Effects and Promotes Autophagy-Mediated Cancer Cell Dormancy in 3D Ovarian Cancer: Role of miR-1305 and of Its Target ARH-I

**DOI:** 10.3390/cancers14092142

**Published:** 2022-04-25

**Authors:** Andrea Esposito, Alessandra Ferraresi, Amreen Salwa, Chiara Vidoni, Danny N. Dhanasekaran, Ciro Isidoro

**Affiliations:** 1Laboratory of Molecular Pathology, Department of Health Sciences, Università del Piemonte Orientale “A. Avogadro”, Via Solaroli 17, 28100 Novara, Italy; andrea.esposito@uniupo.it (A.E.); alessandra.ferraresi@med.uniupo.it (A.F.); salwa.amreen@uniupo.it (A.S.); chiara.vidoni@med.uniupo.it (C.V.); 2Stephenson Cancer Center, The University of Oklahoma Health Sciences Center, Oklahoma City, OK 73104, USA; danny-dhanasekaran@ouhsc.edu

**Keywords:** autophagy, tumor microenvironment, ARH-I, interleukin-6, ovarian cancer, epigenetics, 3D spheroids, DIRAS3, microRNA

## Abstract

**Simple Summary:**

Tumor dormancy is the period during which patients are asymptomatic before recurrence and it is difficult to target pharmacologically. Inflammatory cytokines present in the tumor microenvironment likely contribute to this event. We show that IL-6 inhibits autophagy in ovarian cancer cells via miRNAs downregulation of ARH-I, an effect contrasted by the nutraceutical resveratrol (RV). In detail, RV keeps the cancer cells in an autophagy-mediated dormant-like state contrasting the IL-6 pro-growth activity. Additionally, we show that *ARH-I* (*DIRAS3*) is a *bona fide* target of miR-1305, a novel oncomiRNA upregulated by IL-6 and downregulated by RV. Our findings have a translational impact since we observed that patients with high *DIRAS3* expression have a better outcome, and this correlates with *MIR1305* downregulation. Overall, maintaining a permanent cell dormancy by the chronic administration of RV might be considered a therapeutic option to prevent the “awakening” of cancer cells in response to a permissive microenvironment.

**Abstract:**

Tumor dormancy is the extended period during which patients are asymptomatic before recurrence, and it represents a difficult phenomenon to target pharmacologically. The relapse of tumors, for instance arising from the interruption of dormant metastases, is frequently observed in ovarian cancer patients and determines poor survival. Inflammatory cytokines present in the tumor microenvironment likely contribute to such events. Cancer cell dormancy and autophagy are interconnected at the molecular level through ARH-I (DIRAS3) and BECLIN-1, two tumor suppressors often dysregulated in ovarian cancers. IL-6 disrupts autophagy in ovarian cancer cells via miRNAs downregulation of ARH-I, an effect contrasted by the nutraceutical protein restriction mimetic resveratrol (RV). By using three ovarian cancer cell lines with different genetic background in 2D and 3D models, the latter mimicking the growth of peritoneal metastases, we show that RV keeps the cancer cells in a dormant-like quiescent state contrasting the IL-6 growth-promoting activity. Mechanistically, this effect is mediated by BECLIN-1-dependent autophagy and relies on the availability of ARH-I. We also show that *ARH-I* (*DIRAS3*) is a *bona fide* target of miR-1305, a novel oncomiRNA upregulated by IL-6 and downregulated by RV. Clinically relevant, bioinformatic analysis of a transcriptomic database showed that the high expression of *DIRAS3* and *MAP1LC3B* mRNAs together with that of *CDKN1A*, directing a cellular dormant phenotype, predicts better overall survival in ovarian cancer patients, and this correlates with *MIR1305* downregulation. The possibility of maintaining a permanent cell dormancy in ovarian cancer by the chronic administration of RV should be considered as a therapeutic option to prevent the “awakening” of cancer cells in response to a permissive microenvironment, thus limiting the risk of tumor relapse and metastasis.

## 1. Introduction

Tumor dormancy corresponds to the extended period during which patients are asymptomatic before relapse and metastasis occurrence [1]. Tumor mass dormancy refers to a condition of equilibrium between cell proliferation and cell death that maintains the volume of the tumor over a long time. This quasi-static condition lasts until the tumor is starved of nutrients, growth factors, and oxygen due to poor vascularization (angiogenic tumor dormancy) or tumor cell proliferation is balanced by immune killing (immune-mediated tumor dormancy) [2]. Cancer-cell dormancy, which also determines a tumor dormancy state, refers to the condition in which the totality (or a large part) of the cancer cells is in a quiescent, non-proliferative state characterized by low metabolic activity [3]. Dormant cancer cells are drug-resistant and are responsible for cancer relapse once they regain the ability to grow [4]. However, keeping cancer cells in a dormant state for decades could be a valuable strategy to preserve patients’ life. This is, for instance, believed to occur in ER-positive breast cancer patients under tamoxifen therapy (reviewed in [2]). 

Metastatic dormant cancer cells may remain silent for many years [4]. For instance, in ovarian cancer patients, relapse may present after ten years and this has been attributed to the slow outgrowth of cancer cells adherent to the peritoneum [5], the majority of which likely have been dormant for such a long period. Ovarian cancer is the main cause of mortality for gynecological malignancies [6], and the persistence of metastatic, drug-resistant, dormant cancer cells is one of the most important factors contributing to progression and poor patients’ outcome [7,8,9].

Tumor dormancy is considered a fundamental adaptive response to stress conditions within the tumor microenvironment (TME) [10]. Accordingly, a restrictive or permissive TME induces cancer cells to enter or exit dormancy, respectively [3]. The survival of cancer cells in the dormant state within a TME hypo-vascularized and lacking in nutrients and oxygen depends on autophagy [11]. The crosstalk between cancer cells and stroma finely orchestrates ovarian cell dormancy via soluble factors (e.g., cytokines, chemokines, growth factors, etc.) and epigenetic mechanisms that modulate autophagy in cancer cells [1,12,13,14]. Cancer-associated fibroblasts (CAFs)-derived inflammatory cytokines such as IL-6 and IL-8 have been shown to promote ovarian cancer cell proliferation and motility through the downregulation of autophagy in the migrating cells [15,16,17,18,19]. Particularly, IL-6 has been shown to induce a long-lasting downregulation of autophagy via miRNAs targeting ARH-I, an interactor and activator of BECLIN-1-dependent autophagy [15]. Of note, *ARH-I* (aka *DIRAS3*) is a tumor suppressor gene maternally imprinted and found to be mono-allelically silenced in more than 60% of ovarian cancers [20]. ARH-I has been shown to be responsible for controlling the cancer cell dormant state through the modulation of autophagy in ovarian cancer cells facing nutrient deprivation [21]. Autophagy is a metabolic process for lysosome-driven macromolecular degradation that helps the cell to overcome damaging injuries (including oxidative damages) and to survive under stressful situations such as starvation [22]. We shall not describe in detail the autophagy machinery and regulation, as this can be found in an excellent review [23].

We have shown that the nutraceutical resveratrol (RV) is a good caloric restriction mimetic that can induce autophagy in ovarian cancer cells even stronger than amino acids and serum starvation [24]. Of relevance for the aim of the present work, we have previously shown that RV counteracts the IL-6-induced inhibition of autophagy in ovarian cancer cells by rescuing the expression of ARH-I, an effect linked to the opposite regulation of miRNAs targeting the mRNA of this protein [15].

Here, we provide evidence that the chronic administration of RV induces a cell quiescent state with features of cell dormancy in ovarian cancer cells and, more interestingly, that this dormant state persists also when the cells are challenged with IL-6. Mechanistically, IL-6 and RV oppositely regulate autophagy-dependent ovarian cancer cell proliferation and dormancy through the modulation of miR-1305, which is proven to *bona fide* target ARH-I. The bioinformatic interrogation of a clinical transcriptomic database revealed that patients with a high expression of *DIRAS3* (*ARH-I*) and *MAP1LC3B* (coding for the autophagosomal protein LC3) together with the upregulation of *CDKN1A* (coding for the cyclin-dependent kinase inhibitor p21 protein) have better overall survival, and this correlates with *MIR1305* downregulation.

Taken together, these findings have a translational relevance supporting RV as an adjuvant therapeutic for a “sleeping-inducing strategy” for maintaining cancer cells in a persistent dormant state, also in the context of an inflammatory TME.

## 2. Materials and Methods

### 2.1. Cell Culture

The NIH-OVCAR-3 cell line was obtained from ATCC (cod. HTB-161; American Type Culture Collection; Manassas, VA, USA). The OAW42 cell line was obtained from ECACC (cod. 85073102; European Collection of Authenticated Cell Cultures; Porton Down, Salisbury, UK). The KURAMOCHI cell line was obtained from JCRB Cell Bank (cod. JCRB0098; Japanese Collection of Research Bioresources Cell Bank; Japan). The OVCAR3 and KURAMOCHI human ovarian cancer cell lines were cultured in RPMI 1640 medium (cod. R8758; Sigma-Aldrich, St. Louis, MO, USA) supplemented with 10% heat-inactivated fetal bovine serum (FBS, cod. ECS0180L; Euroclone, Milan, Italy), 1% glutamine (cod. G7513; Sigma-Aldrich), and 1% penicillin/streptomycin (PES, cod. P0781; Sigma-Aldrich).

The OAW42 human ovarian cancer cell line was cultured in Minimum Essential Medium (MEM) (cod. M2279; Sigma-Aldrich) supplemented with 10% heat-inactivated FBS, 1% glutamine, 1% non-essential amino acids (cod. M7145; Sigma-Aldrich), and 1% PES.

The OVCAR3-GFP-LC3 cells were established in our laboratory [15] and maintained in culture conditions as the parental cell line. 

All the cell lines were maintained under standard culture conditions (37 °C, 5% CO_2_).

### 2.2. Reagents

Resveratrol (RV, cod. R5010, Sigma-Aldrich) was dissolved in DMSO and used at different concentrations (100 µM, 20 µM, or 10 µM) as indicated. Interleukin-6 (IL-6, cod. 11340066, Immunotools, Friesoythe, Germany) was dissolved in sterile water and used at a final concentration of 50 ng/mL. Control experiments demonstrated that DMSO (final concentration: 0.01%) had no effect on cell growth and autophagy.

The miR-1305 mimic and anti-miR-1305 (i.e., miR-1305 inhibitor) were purchased from Life Technologies (Carlsbad, CA, USA) and used at a final concentration of 200 pmol/mL.

### 2.3. Antibodies

The following primary antibodies were employed for either immunofluorescence or western blotting: mouse anti-ARH-I (for IF 1:250, for WB 1:1000; cod. ab45768; Abcam, Cambridge, UK), mouse anti-p21 (for IF 1:100, for WB 1:200; cod. sc-817; Santa Cruz Biotechnology, Dallas, TX, USA), rabbit anti-LC3 (1:2000; cod. L7543; Sigma-Aldrich), rabbit anti-p62 (1:500; cod. 8025; Cell Signaling, Danvers, MA, USA), mouse anti-β-actin (1:2000; cod. A5441; Sigma-Aldrich), mouse anti-BECLIN-1 (1:500; cod. 612112; BD Biosciences, Franklin Lakes, NJ, USA), goat anti-BECLIN-1 (1:250; cod. sc-10086; Santa Cruz Biotechnology), rabbit anti-VPS34 (1:500; cod. 4263; Cell signaling), mouse anti-STAT3 (1:500; cod. 9139; Cell Signaling), rabbit anti-phospho (Tyr705) STAT3 (1:1000; cod. 9145; Cell Signaling), rabbit anti-Ki-67 (1:100; cod. HPA001164; Sigma-Aldrich), mouse anti-BCL-2 (1:500; cod. 15071; Cell Signaling), mouse anti-IL-6R (1:1000; cod. AHR0061; Invitrogen, Waltham, MA, USA), rabbit anti-cyclin D1 (1:500; cod. 2978; Cell Signaling), rabbit anti-p38 (1:500; cod. 9212; Cell Signaling), rabbit anti-GAPDH (1:1000, cod. G9545, Sigma Aldrich), and mouse anti-ERK1/2 (1:500; cod. 05-1152; Millipore, Burlington, MA, USA).

### 2.4. 3D Spheroids Proliferation Assay

Cells were seeded at a density of 500,000 cells/petri dish using non-adherent dishes coated with Poly-HEMA (2-hydroxyethylmethacrylate, cod. P3932; Sigma-Aldrich) and cultured as described previously [25]. Spheroid growth was monitored and documented under the phase-contrast microscope. Spheroid dimension was determined with ImageJ software (v. 1.52). Cell proliferation was monitored also by means of DiD staining. Briefly, 1 million cells stained with 1 µM DiD (cod. V22887; Life Technologies) were plated, and once formed, the spheroids were treated as indicated, harvested, and cyto-spotted on glasses. The intensity of DiD retention (which decays along with cell division) was documented under the fluorescence microscope (Leica DMI6000; Leica Microsystems, Wetzlar, Germany) and then quantified with ImageJ software.

### 2.5. Western Blotting

A standard procedure was used for preparing cell homogenates, Western blotting onto a PVDF filter (cod. 162-0177; BioRad, Hercules, CA, USA), and the saturation of the membrane with non-fat dry milk, as detailed in [25]. Filters were incubated with the appropriate primary antibodies overnight at 4 °C, followed by incubation with secondary HRP-conjugated antibodies (goat anti-mouse (cod. 170-6516) or goat anti-rabbit (cod. 170-6515)) for 1 h at room temperature. The bands were detected with Enhanced Chemiluminescence reagents (ECL, cod. NEL105001EA; Perkin Elmer, Waltham, MA, USA) and imaged with the VersaDOC Imaging System (Universal Hood II—S.N. 76S/04219; Biorad, Milan, Italy). For loading control, the filters were re-probed with β-ACTIN or GAPDH. Densitometric analysis was performed with Quantity One software (v. 4.5). Original Western Blot data is shown in Appendix A.

### 2.6. Clonogenic Assay

Cells were seeded into Petri dishes at a density of 10,000 cells/cm^2^ and treated as per the experimental conditions. Substances were re-added to fresh medium daily. Cells were fixed with methanol and stained with 0.5% crystal violet solution [16]. The Petri dishes were extensively washed, dried at room temperature, and photographed. Colony growth was estimated by ImageJ software.

### 2.7. Cell Counting, Doubling Time, and Cell Cycle Analysis

Cells were plated in Petri dishes (5000–10,000 cells/cm^2^) and treated under different conditions. Cells were collected and counted in quadruplicate for each experimental condition. Doubling time (Dt) was calculated using the software Doubling Time Online Calculator (http://www.doubling-time.com/compute.php; accessed on 10 March 2021). Cells were fixed in 70% ethanol, stained with propidium iodide (PI, 50 μg/mL; cod. P4170, Sigma Aldrich), and acquired by using a FACSCalibur (Becton Dickinson, Eysins, Switzerland) flow cytometer. For each sample, a fraction of 5000 events was analyzed. The analysis of the cytofluorimetric data obtained was performed by using Flowing software (v. 2.5.1).

### 2.8. Immunofluorescence

Cells were plated on sterile coverslips at a density of 5000–10,000 cells/cm^2^ and treated or transfected as appropriate. A standard procedure was used for immunofluorescence, as detailed in [25]. The coverslips were incubated overnight at 4 °C with specific primary antibodies and the following day were incubated with dye-conjugated secondary antibodies for 1 h at room temperature. Nuclei were stained with the UV fluorescent dye DAPI (4′,6-diamidino-2-phenylindole). Coverslips were mounted onto glasses using SlowFade reagent (cod. S36936; Invitrogen) and imaged under a fluorescence microscope (Leica DMI6000).

### 2.9. Co-Immunoprecipitation Assay

Cells were plated on Petri dishes (50,000 cells/cm^2^), allowed to adhere, and treated as indicated. Cells were collected with lysis buffer and the protein content was measured by BCA assay. The same amount of protein (500 μg) was incubated with the anti-ARH-I antibody (5 μg), and to capture the immunocomplexes, 50 μL of Sepharose G beads (cod. P3296; Sigma-Aldrich) were added to each sample. Immunocomplexes were then precipitated by centrifugation and eluted with Leammli buffer. Samples were loaded on an SDS-PAGE and immunoblotted with specific antibodies to reveal the presence of ARH-I interactors (BECLIN-1 and VPS34).

### 2.10. Cell Transfection

Cells were plated on coverslips or on Petri dishes depending on the experiment performed. The miRNA-loaded liposomal complexes were prepared in Opti-MEM I Reduced Serum Medium (cod. 11058021, Life Technologies) with 200 pmol of miR-1305 or anti-miR-1305 and Lipofectamine 3000 (cod. L3000-015, Life Technologies). After 6 h of incubation, the medium was replaced with a complete culture medium, which was renewed every 24 h to prevent secondary effects due to starvation.

### 2.11. Assessment of Autophagy in Living Cells Expressing GFP-LC3

OVCAR3-GFP-LC3 cells were seeded on coverslips (2000 cells/cm^2^) and treated depending on the experiment type. Coverslips were washed, mounted, and immediately imaged under the fluorescence microscope (Leica DMI6000) as reported previously [15]. 

### 2.12. Bioinformatic Prediction of miR-1305/ARH-I Interaction

miR-Walk 2.0 (http://mirwalk.umm.uni-heidelberg.de/; accessed on 15 January 2021), a bioinformatic software that includes 12 algorithms (mirWalk, Microt4, miRanda, mirbridge, miRDB, miRMap, miRNAMap, Pictar2, PITA, RNA22, RNAhybrid, and TargetScan), was used to retrieve the predicted interactions between miRNAs and the 3′UTR region of *ARH-I* transcript.

### 2.13. Bioinformatic Analysis of TCGA Clinical Data

Kaplan–Meier curves and correlation studies were conducted by extracting clinical data from the TCGA database (www.portal.gdc.cancer.gov/, accessed on 16 November 2021). The analysis was conducted on the ovarian serous cystadenocarcinoma dataset (TCGA Nature 2011), which accounts for 316 patients. mRNA expression and clinical information (e.g., overall survival) were downloaded from cBioportal.org. Patients were grouped based on the level of mRNA expression and copy number variations (CNV). Low vs. high groups were defined relative to the median expression level of the overall patient cohort.

Scatter plots were employed to represent the correlation between the expression of relevant biomarkers in the patient cohort. Regression was estimated by calculating Spearman’s and Pearson’s correlation coefficients (r) and the relative *p*-values.

Statistical analyses were performed by R (3.6.1 version, The R Foundation for Statistical Computing, Vienna, Austria) and SAS software (9.4. version, SAS Institute Inc., Cary, NC, USA). The log-rank test has been used to determine the statistical significance. The *p*-value ≤ 0.05 was considered significant.

### 2.14. Imaging Acquisition and Analysis

Fluorescence images were acquired by a fluorescence microscope (Leica DMI6000). For each experimental condition, at least three slides were prepared in separate experiments and five to ten microscopic fields, randomly chosen, were imaged by two independent investigators unaware of the treatment. The quantification of fluorescence intensity was performed with ImageJ software (https://imagej.nih.gov/; accessed on 15 April 2018). The representative images of selected fields are shown.

### 2.15. Statistical Analysis

Statistical analysis was performed with GraphPad Prism 5.0 software. Bonferroni’s multiple comparison test after one-way ANOVA analysis (unpaired, two-tailed) was employed. Significance was considered as follows: **** *p*< 0.0001; *** *p* < 0.001; ** *p* < 0.01; * *p* < 0.05. All data are reported as average ± S.D.

## 3. Results

### 3.1. Resveratrol Contrasts IL-6 Stimulation of 3D Tumor Spheroid Growth

To evaluate the growth of ovarian cancer cells in a condition that mimics, at best, the tumor growing in the peritoneum, we performed a 3D spheroid-forming assay [26]. Three different ovarian cancer cell lines, namely OVCAR3, OAW42, and KURAMOCHI, as representatives of ovarian cancers with different patterns of oncogenes, tumor suppressor genes, and microRNAs expression and different metabolic alterations and extensions of malignant features [27,28,29] were employed. Cells were chronically treated with interleukin-6 (IL-6) to mimic in vitro the pro-inflammatory TME found in ovarian cancer patients [13]. After 5 days of treatment with IL-6, the cultures were supplemented with resveratrol (10 µM RV), a caloric restriction mimetic and autophagy inducer [24]. 

Compared to the control, the chronic exposure to IL-6 significantly increased the 3D spheroid dimension as early as day 2 (Figure 1). The addition of RV on day 5 resulted in tumor spheroid growth arrest. Strikingly, this effect was observed also in IL-6-treated 3D spheroid cultures and lasted for the subsequent five days. The average area of the spheroids at day 10 increased by up to 3-fold in IL-6-treated cultures, while it was about half that of the controls in the RV-supplemented cultures. These effects were more pronounced, in order, in OVCAR3, OAW42, and KURAMOCHI cells.

We investigated the molecular pathways involved in the observed effects, focusing on the IL-6R (IL-6 receptor)/STAT3 pathway. The expression of IL-6R and the phosphorylation of STAT3 were assayed on homogenates of OVCAR3, OAW42, and KURAMOCHI 3D spheroids cultured for 10 days as above (Figure 2A). The treatment with IL-6 resulted in increased IL-6R expression along with STAT3 Tyr705 phosphorylation, and this effect was canceled by the addition of 10 μM RV on day 5 (Figure 2B,C,E,F,H,I). It has been reported that STAT3 can be sequestered in the cytosol through binding with ARH-I, a maternally imprinted tumor suppressor involved in the regulation of ovarian cancer cell growth and dormancy [21]. We previously reported that in ovarian cancer cells grown as 2D, the level of ARH-I is reduced by IL-6 and increased by RV, respectively [15]. Therefore, we evaluated the expression of ARH-I in the 3D spheroid cultures. While IL-6 downregulated the expression of ARH-I, the addition of RV on day 5 rescued and increased its expression (Figure 2D,G,J). Notably, the highest upregulation of ARH-I (up to 20-fold) in parallel with the more pronounced IL-6-mediated STAT3 activation (about 5-fold) was recorded in OVCAR3 3D spheroids, consistent with the effects on the growth reported in Figure 1. This prompted us to investigate further in-depth the effects of RV on this cell line.

### 3.2. Resveratrol Induces a G0/G1 Cell Cycle Arrest in OVCAR3 Cells Exposed to IL-6

To evaluate cell proliferation in OVCAR3 spheroids, we employed DiD staining, a fluorescent dye that rapidly and stably integrates into the phospholipid cell membrane and whose signal is diluted along with cell duplication. OVCAR3 spheroids chronically exposed to IL-6 (for ten days, as for the experiments described above) showed decreased DiD fluorescence signals compared to the control, indicative of the occurred cell proliferation (Figure 3A,B). The DiD signal was rescued in the spheroids that had been co-treated with RV from day 5, indicating that cell division was impaired (Figure 3A,B). Consistently, at day 10, RV contrasted the ERK1/2-mediated proliferation signaling induced by IL-6 and concomitantly triggered the expression of the cell cycle inhibitor p21 (Figure 3C).

The colony assay confirmed that the supplementation of RV was able to arrest the growth of IL-6-stimulated OVCAR3 cells (Figure 3D), in agreement with the increased doubling time (Figure 3E). The cytofluorimetric analysis of the cell cycle showed that after five days of exposure to IL-6, the amount of OVCAR3 cells in the S phase was doubled, while it remained like that of controls in the subsequent five days when co-treated with RV (Figure 3F). Immunofluorescence staining showed that IL-6 stimulated the expression of nuclear Ki-67 and of cytoplasmic cyclin D1 (consistent with the transition to the S phase), along with the reduced expression of ARH-I, and that the co-incubation with RV in the last five days led to decreased expression of both Ki-67 and cyclin D1 (the latter appears relocated in the nucleus) and concomitant increase of ARH-I and p21 (Figure 3G). These data are consistent with cytofluorometric data showing an accumulation of cells in the G0/G1 phase (shown in panel F) and are suggestive of a dormant phenotype in the culture exposed to RV.

### 3.3. Ovarian Cancer Spheroids Pre-Treated with Resveratrol Do Not Rescue the Growth upon Challenge with IL-6

Next, we tested whether RV could induce a long-lasting dormant state in OVCAR3 cells that could prevent IL-6 re-growth stimulation. To this end, we set up a protocol in which OVCAR3 spheroids were chronically exposed to RV, progressively decreasing its concentration to avoid cytotoxicity yet at the same time keeping the living cells in a dormant-quiescent state (Figure 4A). On day 5 and for the next five days we challenged the culture with IL-6 still in the presence or, alternatively, absence of RV. These experimental conditions mimic, at best, the occurrence of an accidental recrudescence of inflammation at the metastatic site in a tumor-bearer subjected to chronic RV treatment or who interrupted the therapy.

In line with the results above, chronic treatment with RV impaired tumor growth, as suggested by the fact that the spheroids dimensions remained constant during the time while it progressively increased in the controls. Based on the tumor spheroids dimensions, the challenge with IL-6 on day 5 did not result in an obvious rescue of cell proliferation in the subsequent five days, in either the cultures where RV had been withdrawn or was continuously present (Figure 4B,C).

Next, we assayed the IL-6R/STAT3 signaling pathway level in the OVCAR3 cells grown as 3D and treated as above (Figure 5A). The chronic (ten days) treatment with RV reduced by approximately 5-fold the levels of IL-6R and nearly completely abrogated the phosphorylation (Tyr705) of STAT3 (Figure 5B,C). The addition of IL-6 only slightly changed this pattern, indicating that pre-treatment with RV effectively protected the cells from being challenged by the inflammatory cytokine. Additionally, ARH-I expression remained significantly elevated in RV-treated cells (about 3–4-times higher than the control) even upon challenge with IL-6 (Figure 5D), suggesting RV elicited long-lasting benefits, possibly due to epigenetic mechanisms.

To determine whether the pre-treatment with RV could turn the cells into a quiescent, dormant-like state, we performed the same analyses as for the experiments described in Figure 3. The DiD staining of OVCAR3 cells grown as 3D and treated for ten days as described in Figure 4 indicated that IL-6 could not trigger cell proliferation when the culture was pre-treated or co-treated with RV (Figure 6A,B). The tumor spheroids pre-treated with RV showed reduced levels of ERK1/2 expression (about 50% less than in the control) and increased levels of p38 and p21 (about 4-fold and 8-fold, respectively), and this same pattern remained essentially unchanged in the spheroids that were challenged from day 5 to day 10 with IL-6, whether RV was removed or still present (Figure 6C). The colony assay further confirmed the inability of IL-6 to restart cell proliferation in the cultures pre-exposed to RV, regardless of whether the latter was removed or maintained in the medium (Figure 6D). The doubling time was consistent with the above data, showing that RV could prolong it by 2-fold and that IL-6 could not shorten it (Figure 6E). In line with these results, the cancer cells chronically treated with RV accumulate in the G0/G1 phase and do not re-enter the cell cycle upon the addition of IL-6 from day 5, even in the culture where on day 5 RV was withdrawn (Figure 6F). Notably, upon challenging with IL-6, a small percentage of cells entered the S phase, yet these cells were unable to complete the cell cycle, as indicated by the fact that the proportion of cells in the G2/M phase remained the same as in RV-treated cells (Figure 6F). Finally, we evaluated the expression of markers associated with the quiescent, dormant-like phenotype. Immunofluorescence double-staining for ARH-I/Ki-67 and for p21/cyclin D1 showed that RV (5 days pre-treatment or 5 days pre-treatment plus 5 days co-treatment with IL-6) upregulated the expression of ARH-I and p21 and concomitantly downregulated the expression of Ki-67 and cyclin D1, and this pattern was not altered by the challenge with IL-6 (Figure 6G).

### 3.4. Resveratrol Promotes an ARH-I-Mediated Autophagy in Contrast to IL-6

Previous studies have shown that the overexpression of DIRAS3/ARH-I may drive ovarian cancer cells into a dormant state through the induction of autophagy [30]. We checked whether this also occurred in OVCAR3 cells exposed to RV. First, we assessed autophagy in OVCAR3 tumor spheroids cultivated for ten days as per the experimental workflow presented in Figure 1. The Western blotting of p62/SQSTM1, a surrogate marker of cargo degradation [31], accumulated in IL-6-treated cells, and this accumulation was contrasted by co-treatment with RV from day 5, indicating that IL-6 slowed down while RV stimulated the autophagy flux (Figure 7A). This interpretation is supported by the reduced conversion of LC3-I into LC3-II (indicative of autophagosome formation) in IL-6-treated spheroids that is instead promoted by RV (Figure 7A). A similar protocol of treatment in the 2D cultures of GFP-LC3 expressing OVCAR3 showed a diffuse fluorescence staining in IL-6-treated cells, indicating that LC3 remained in its cytosolic form, and an increased number of fluorescence puncta, indicative of vacuolar relocation of LC3, in RV-treated cells (Figure 7B).

Next, we performed the same analyses in OVCAR3 cells treated for ten days as per the experimental protocol illustrated in Figure 4. Again, we observed that p62 expression decreased while LC3-II increased over LC3-I in tumor spheroids exposed to RV for either ten days or only the first five days, indicating that autophagy flux was stimulated, and this effect contrasted the inhibitory action of IL-6 (Figure 7C). This interpretation is further supported in the 2D cultures of GFP-LC3-expressing OVCAR3 cells where fluorescence puncta (associated with autophagic vacuoles) are clearly evident in all conditions with RV (Figure 7D).

ARH-I has been shown to trigger autophagy through displacing BCL-2 and directly interacting with BECLIN-1, thus promoting the autophagosome initiation complex [20]. Thus, we asked whether RV could favor such intermolecular interactions and if this effect could occur also in the presence of IL-6 and persist even after its withdrawal from the culture. To this end, we performed a co-immunoprecipitation assay to identify the main partners of ARH-I involved in the autophagy interactome in OVCAR3 cells incubated for 24 h with either RV or IL-6 or both. As shown in Figure 7E, RV increased while IL-6 reduced the amounts of BECLIN-1 and VPS34 (aka PI3KC3) interacting with ARH-I. This data was further supported by the immunofluorescence co-labeling of BECLIN-1 with either ARH-I or BCL-2. The images in Figure 7F demonstrate that IL-6 promotes BECLIN-1–BCL-2 interaction whereas RV displaces BCL-2 and promotes the interaction of BECLIN-1 with ARH-I, consistent with co-immunoprecipitation data.

### 3.5. miR-1305 Downregulates ARH-I and Induces Ovarian Cancer Cell Proliferation Mimicking IL-6 Effects

In a separate study, we analyzed the microRNome in OVCAR3 cells and found that IL-6 and RV oppositely modulated the expression of six miRNAs potentially targeting ARH-I [15]. The in silico analysis with the miRWalk software indicated that miR-1305 was the most promising candidate since 5 algorithms out of 12 confirmed *ARH-I* as its target (Figure 8A). Consistently, the probability of interaction between miR-1305 and the 3′UTR of *ARH-I* is statistically significant (*p* = 0.0393) (Figure 8B). To validate this prediction at the protein level, we monitored by Western blotting the expression of ARH-I in miR-1305-transfected OVCAR3 cells for up to 72 h. As shown in Figure 8C, much like IL-6, miR-1305 caused a progressive decrease of ARH-I protein in transfected cells, lowering the level by approximately 6-fold compared to the untransfected counterpart after 72 h. Then, it was mandatory to determine whether miR-1305 could resemble IL-6 functionally. To this end, we assessed the proliferation rate, colony formation rate, and cell cycle distribution of the cells transfected with miR-1305 or with anti-miR-1305, the latter being introduced as a counterpart possibly resembling the treatment with RV. Based on the DiD dye retention test (Figure 8D), on the third day of culture, the untransfected control cells contained one-sixth of the initial fluorescence, indicating the occurrence of cell duplication. By this time, DiD fluorescence was halved in miR-1305-transfected cells while it was doubled in anti-miR-1305-transfected cells compared to the controls (Figure 8E). It is remarkable that anti-miR-1305, which sponges endogenous miR-1305, could slow down the proliferation rate, acting much like RV.

To further support this finding, we moved to the colony assay. Figure 8F shows that miR-1305 increases the efficiency of colony formation while the transfection with miR-1305 antagonist results in fewer colonies compared to that of the control. Additionally, we found that miR-1305-transfected cells exhibited an increased accumulation in the G2/M phase in parallel with a reduced accumulation in the G0/G1 phase as compared to the profile of control cells, consistent with an increased proliferation rate. In contrast, the cells transfected with anti-miR-1305 displayed a cytofluorimetric profile like the non-transfected cells (Figure 8G). To see if these effects were mechanistically linked with ARH-I expression, we performed the immunofluorescence co-staining of this protein with Ki-67 in the transfected cells. The images shown in Figure 8H demonstrate that compared to controls, the cells transfected with the miR-1305 mimic express low levels of ARH-I along with high levels of Ki-67, whereas the cells transfected with the anti-miR-1305 display the opposite pattern. To definitively verify that miR-1305 could mimic IL-6 in inducing a quiescent/dormant-like state in the cells, we assessed by Western blotting the expression of relevant biochemical markers (Figure 8I). Compared to the pattern of expression in the untransfected control, the transfection with miR-1305 resulted in the upregulation of ERK1/2 along with the downregulation of p38, and ARH-I, whereas the transfection with anti-miR-1305 resulted in the opposite regulation of these proteins. Remarkably, anti-miR-1305 greatly upregulated cell dormancy-associated markers p38 (about 2-fold), ARH-I (2.5-fold), and p21 (3-fold).

Finally, we wanted to prove that the phenotypic effects of miR-1305 and anti-miR-1305, respectively mimicking IL-6 and RV, were indeed linked to autophagy modulation. First, we checked by co-immunoprecipitation the formation of the pro-autophagic interactome in the transfected cells. Compared to control cells, the amount of BECLIN-1 and VPS34 bound to ARH-I is reduced in miR-1305-transfected cells, while it is increased in anti-miR-1305-transfected cells (Figure 9A). Consistently, miR-1305 downregulates while anti-miR-1305 upregulates autophagy, as indicated by the changes in p62 and in the LC3-II/LC3-I ratio (Figure 9B). Notably, anti-miR-1305 abrogates IL-6-induced effects on ARH-I-mediated autophagy (Figure 9C). Accordingly, in GFP-LC3-expressing cells, anti-miR1305 promotes the formation of LC3 puncta, indicative of autophagy induction, while both IL-6 and miR-1305 determine a diffused cytosolic fluorescence, indicative of autophagy inhibition (Figure 9D).

### 3.6. High Expression of DIRAS3 and MAP1LC3B Correlates with CDKN1A and Predicts Good Prognosis in Ovarian Cancer Patients

Finally, we addressed the translational relevance of autophagy-regulated cancer cell dormancy in the clinical outcomes of ovarian cancer patients. We interrogated the TCGA database to evaluate whether the *DIRAS3*-related signature correlates with the autophagy-mediated quiescent state of the tumor in patients’ samples. To this end, we searched for the correlation between the expression of the relevant genes, namely *DIRAS3* (coding for ARH-I), *MAP1LC3B* (coding for the autophagic marker LC3), and *CDKN1A* (coding for the cyclin-dependent kinase inhibitor p21). We found that the *DIRAS3* mRNA level positively correlates with the *CDKN1A* mRNA level (Figure 10A) and the latter correlates with *MAP1LC3B* mRNA expression (Figure 10B). We also compared the survival outcome of patients bearing differential expressions of *DIRAS3* and *MAP1LC3B*. First, we found that these two genes are positively associated (Figure 10C). Most importantly, the patients with high *DIRAS3* expression together with *MAP1LC3B* upregulation (that indicates active ongoing autophagy) had a significantly better prognosis (*p* = 0.0579) than those belonging to the other groups (Figure 10D).

Next, we examined whether miR-1305 could be implicated in the clinical outcome of those patients. We stratified the patients based on the expression of *DIRAS3* and *MIR1305* and checked for their clinical outcomes. Forty-two patients presented the shallow deletion of *MIR1305* and 273 patients presented no alterations in the copy number variation (CNV) of *MIR1305*. We found that *MIR1305* levels were inversely correlated to *DIRAS3* mRNA (Figure 10E) and that the patients bearing a shallow deletion of *MIR1305* together with *DIRAS3* upregulation exhibited a longer overall survival than the patients with no alteration in *MIR1305* CNV (Figure 10F).

## 4. Discussion

Despite the huge progress recently made in the treatment of ovarian cancer, too many patients still experience tumor relapse, thus resulting in poor survival outcomes [1,8,32]. Tumor relapse has been linked to the interruption of tumor dormancy, a condition in which the tumor mass remains constant for a long time due to the balance between cell growth and immune-mediated cell death (immune-dormancy), lack of vascularization (angiogenic dormancy), or cell quiescence (cancer cell dormancy) [2]. The development of metastasis along with the high heterogeneity of the malignant lesions, the intricate crosstalk of signaling pathways, and the resistance to therapy make tumor dormancy a difficult process to control from a pharmacological point of view [33].

Cancer cell dormancy is an important response of adaptation to microenvironmental stress conditions, resulting from metabolic crosstalk, involving cytokines, chemokines, growth factors, metabolites, and non-coding RNAs, in which autophagy plays a fundamental role [12,34]. Autophagy is a dynamic catabolic lysosome-mediated process that ensures the macromolecular turnover and cellular quality control crucial for homeostasis [35]. Cancer cell dormancy and autophagy are interconnected at molecular level through ARH-I (also known as DIRAS3), a maternally imprinted oncosuppressor that is expressed by normal epithelial cells while it is downregulated in more than 60% of ovarian cancer cases [30]. ARH-I can sequester STAT3, which mediates IL-6-transcriptional effects associated with tumor aggressiveness [36]. ARH-I can displace BCL-2 and bind to BECLIN-1 to promote the formation of the autophagic initiation complex together with VPS34/PI3KC3 [20]. Here, we investigated the mechanisms involved in the reciprocal regulation of autophagy and cell dormancy in ovarian cancer cells challenged with IL-6 to mimic the pro-inflammatory tumor microenvironment.

We used the 3D spheroid culture model that, in vitro, resembles the growth of ovarian cancer cells disseminated in the peritoneum [37,38].

We found that IL-6 promotes the growth of 3D spheroids, and this effect is associated with the downregulation of ARH-I expression and the inhibition of autophagy. ARH-I is expressed in normal epithelial cells, while it is downregulated or lost in more than 60% of ovarian cancer cases due to loss of heterozygosity, DNA methylation, transcriptional regulation, and shortened mRNA half-life but also via epigenetic mechanisms involving non-coding RNAs [30]. For example, the upregulation of specific miRNAs, specifically miR-221 and miR-222, lead to ARH-I downregulation, supporting prostate cancer tumorigenesis by the promotion of cell proliferation and invasiveness [39].

In a previous study, we reported that resveratrol (RV; 3,4,5-trihydroxy-trans-stilbene) can counteract the IL-6-induced downregulation of ARH-I [15]. In that study, we identified a signature of six miRNAs modulated in an opposite manner by IL-6 and RV (upregulated and downregulated, respectively) that have ARH-I as the predicted target [15].

The bioinformatic analysis revealed that miR-1305 was the best candidate with the highest probability to bind the 3′UTR of *DIRAS3*. MiR-1305 plays a complex role in carcinogenesis. Many reports indicate that it shows tumor-suppressive properties by dampening cell proliferation and invasion and sensitizing cancer cells to cell death [40,41,42]. On the other hand, it acts as oncomiRNA by promoting cancer stem cell-like phenotypes and correlates with the poor prognosis of multiple myeloma patients [43]. Additionally, another study shows that the overexpression of miR-1305 induced the differentiation of pluripotent stem cells by promoting G1/S transition, while its downregulation facilitated the maintenance of quiescence and increased cell survival [44]. Here, we report that miR-1305 exerts an oncogenic role in ovarian cancer cells. We observed that ovarian cancer patients bearing a shallow deletion of *MIR1305* together with high *DIRAS3* expression display a significantly better overall survival compared to that observed in the group of patients with no CNV alterations (possibly expressing high *MIR1305*). Accordingly, we found that miR-1305 mimic-transfected cells exhibit an enhanced proliferative capability similar to that observed in IL-6-treated cells, while the downregulation of endogenous miR-1305 (either by RV treatment or by sponging it with the miRNA inhibitor) results in the acquisition of ARH-I-mediated dormant phenotype in parallel with the restoration of autophagy. In agreement with these findings, we found that miR-1305 inhibits the recruitment and activation of the autophagy interactome, resembling the effect of IL-6, while anti-miR-1305 promotes the formation of the autophagic interactome, resembling the effects of RV.

Of clinical relevance, we found that ovarian cancer patients bearing a low level of *MIR1305* and high expression of *DIRAS3* have a longer overall survival compared to patients in which the CNV of *MIR1305* was unaltered. Moreover, the mRNA of *DIRAS3* correlated with that of *MAP1LC3B* and the latter with that of *CDKN1A*. These bioinformatic data agree with the functional data showing that the rescue of ARH-I by RV is associated with increased autophagy (i.e., LC3 processing) and cell quiescence (witnessed by the increased expression of p21). Our data agree with a report from Bast’s group, showing that the high expression of ARH-I is associated with the increased expression of p21WAF1/CIP1 and prolonged progression-free survival [45].

Moreover, RV has recently attracted the attention of many researchers thanks to its several tumor-suppressive properties [46,47,48]. In this study, we found that chronic treatment with RV could also prevent the IL-6-induced resumption of 3D spheroids proliferation, thus reducing the risk associated with relapse and metastasis of the “awakened” clones.

We also provide evidence that the RV-promoted switch toward quiescence was sustained by negative feedback on the IL-6R/STAT3 pathway, which has been associated with tumor growth, angiogenesis, chemoresistance, and immune suppression [49]. This finding has an important translational relevance since IL-6R is frequently overexpressed in ovarian tumors and has been shown to correlate with a poor prognosis in ovarian cancer patients [50].

## 5. Conclusions

To date, it remains to be defined which is the most suitable strategy to target tumor dormancy. The current therapeutic approaches include (i) maintaining cancer cells in a chronic dormant state, by avoiding their “awakening” and the consequent risk of relapse and metastasis; (ii) inducing the re-activation of dormant cancer cells and targeting them with conventional anti-proliferative drugs; and (iii) directly killing dormant cells by pharmacologically inhibiting their survival mechanisms [51]. The downregulation of dormancy-associated miRNAs represents a crucial regulation pattern associated with the switch from quiescence to proliferation [52]. Thus, playing with the epigenetic regulation of dormancy could be a feasible strategy, provided an epigenetic modulator is available. RV may act as an epigenetic regulator and a potent autophagy inducer capable of ameliorating several aspects of the malignant phenotype of cancer cells [46,53,54]. Here, we demonstrated that the chronic intake of RV could serve this function. To be noted, RV counteracted the IL-6-induced proliferation in three different ovarian cancer cell lines, suggesting that the mechanism involved is independent of the genetic background of oncogenes and oncosuppressors. Here, we demonstrated that RV and IL-6 modulate the dormant phenotype through the modulation of ARH-I-dependent autophagy. Mechanistically, RV and IL-6 modulate miR1305, which targets ARH-I. Based on these results, we propose miR-1305 as a novel oncomiRNA. Thus, RV could represent a novel therapeutic tool for an epigenetic “sleeping strategy” for counteracting tumor growth or preventing the “awakening” of cancer cells within a pro-inflammatory microenvironment, thus limiting the risk of tumor recurrence and metastasis.

## Figures and Tables

**Figure 1 cancers-14-02142-f001:**
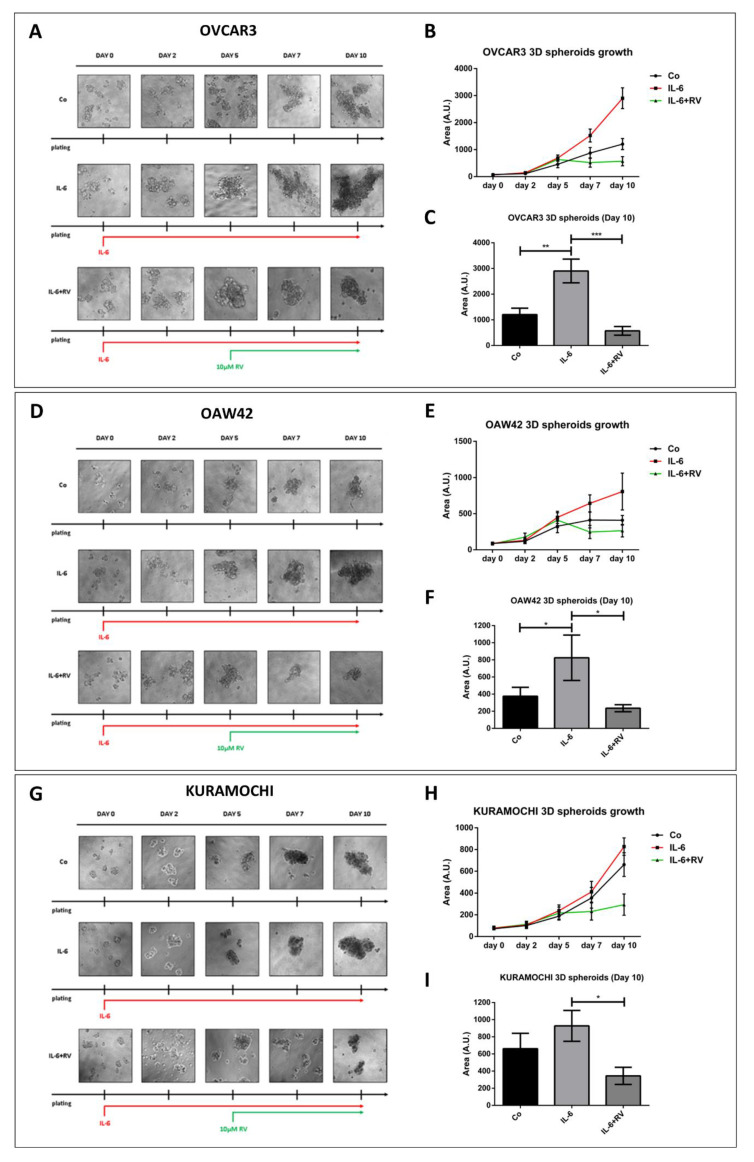
Resveratrol contrasts the IL-6-induced growth of 3D ovarian cancer cell spheroids. OVCAR3, OAW42, and KURAMOCHI cells were grown as 3D spheroids on non-adherent Petri dishes. On day 0, cells were incubated with 50 ng/mL IL-6 and re-treated on days 2, 5, and 7. 10 μM RV was added on days 5 and 7 in combination with IL-6. 3D spheroid dimensions were determined with ImageJ software. Data represent the average ± S.D. calculated for three different fields per condition in three separate experiments. (**A**,**D**,**G**) Time-course phase-contrast photos of OVCAR3, OAW42, and KURAMOCHI 3D spheroids. (**B**,**E**,**H**) Growth curve representing the 3D spheroid dimensions of OVCAR3, OAW42, and KURAMOCHI during the experimental timeline. (**C**,**F**,**I**) Quantification of 3D area on day 10 of OVCAR3, OAW42, and KURAMOCHI spheroids. Statistical analysis was performed by using GraphPad Prism 5.0 software. Bonferroni’s multiple comparison test after one-way ANOVA was employed. Significance was considered as follow: *** *p* < 0.001; ** *p* < 0.01; * *p* < 0.05.

**Figure 2 cancers-14-02142-f002:**
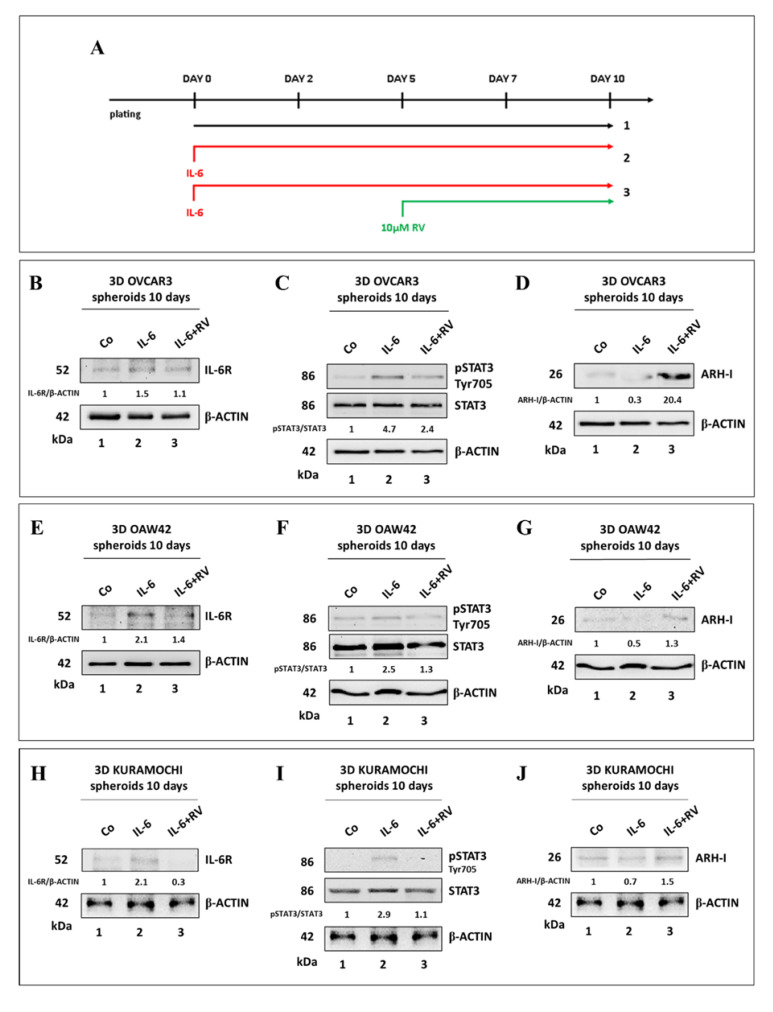
Resveratrol counteracts IL-6-induced IL-6R/STAT3 signaling. (**A**) Experimental timeline of 3D spheroids-forming assay. (**B**,**E**,**H**) Western blotting showing the expression of IL-6R in OVCAR3, OAW42, and KURAMOCHI 3D spheroids. (**C**,**F**,**I**) Western blotting showing the expression of pSTAT3 (Tyr705)/STAT3 in OVCAR3, OAW42, and KURAMOCHI 3D spheroids. (**D**,**G**,**J**) Western Blotting showing the expression of ARH-I in OVCAR3, OAW42, and KURAMOCHI 3D spheroids. The densitometric analysis of the Western blotting representative of three independent experiments is included.

**Figure 3 cancers-14-02142-f003:**
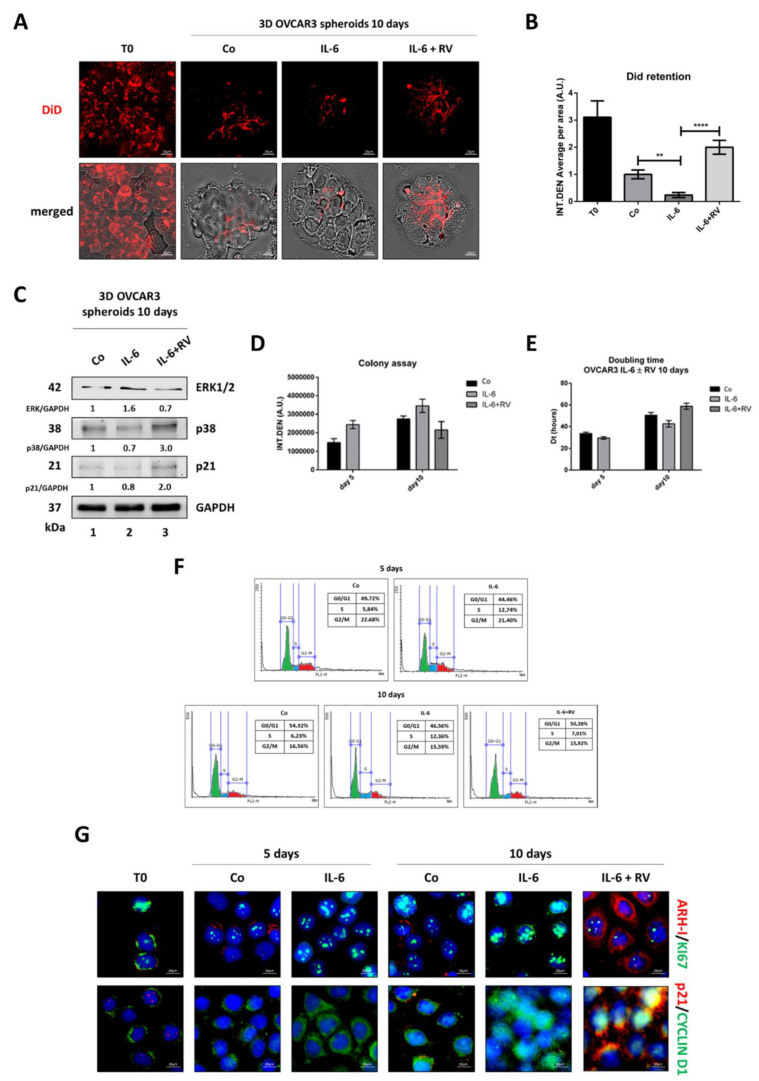
Resveratrol induces dormant-like cell cycle arrest in ovarian cancer spheroids exposed to IL-6. A-B-C. OVCAR3 cells were grown as 3D spheroids and treated following the same timeline as Figure 2. (**A**) The assessment of cell proliferation rate of OVCAR3 3D spheroids. On day 10, DiD-stained 3D spheroids were cytospotted on glass slides and immediately observed under a fluorescence microscope. Scale bar = 20 µm; magnification = 63×. (**B**) Quantification of DiD dye retention performed by ImageJ software. Data ± S.D. represent three independent replicates. Statistical analysis was performed by using GraphPad Prism 5.0 software. Bonferroni’s multiple comparison test after one-way ANOVA was employed. Significance was considered as follows: **** *p* < 0.0001; ** *p* < 0.01. (**C**) Western blotting for the expression of the proliferation/quiescence markers (ERK1/2, p38, and p21) in 3D OVCAR3 spheroids collected on day 10. The densitometric analysis of Western blotting representative of three independent experiments is included. (**D**–**G**). OVCAR3 cells were cultured in 2D and treated as above. (**D**) Colony formation assay calculated on day 5 and day 10. Data ± S.D. are representative of three independent replicates. (**E**) Doubling time calculated on day 5 and day 10. Data ± S.D. are representative of three independent replicates. (**F**) Cell cycle analysis performed on day 5 and day 10 by using a FacScan flow cytometer (**G**) Immunofluorescence double-staining performed on day 5 and day 10 for ARH-I (red)–Ki-67 (green) and for p21 (red)–cyclin D1 (green). Scale bar = 20 µm; magnification = 63×.

**Figure 4 cancers-14-02142-f004:**
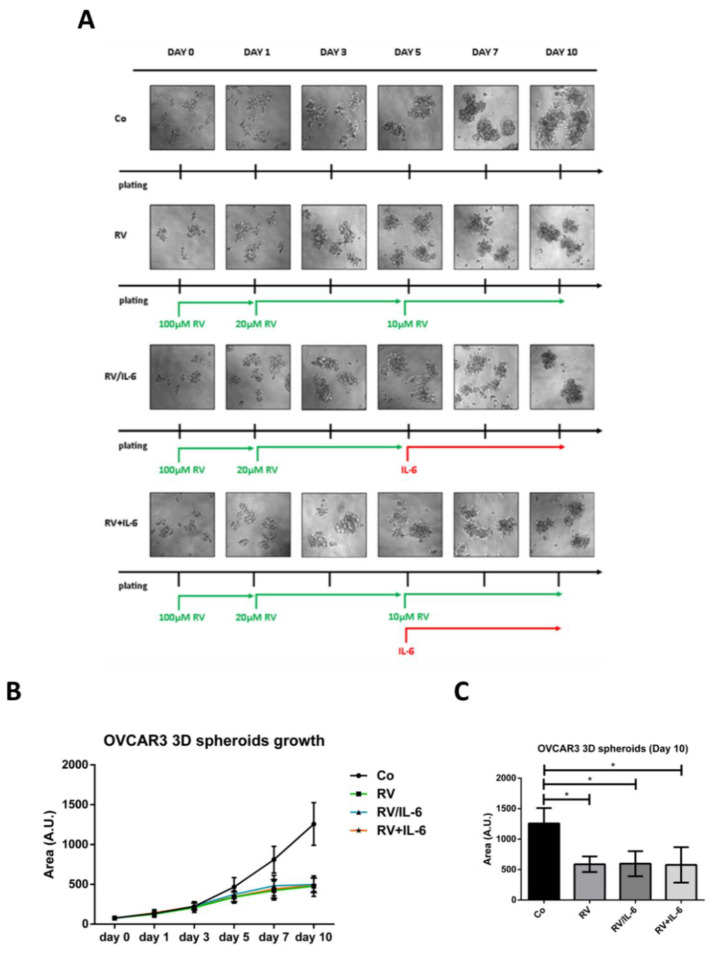
Resveratrol maintains constant the dimensions of 3D ovarian cancer spheroids even in the presence of a pro-inflammatory stimulus. OVCAR3 cells were grown as 3D spheroids. On day 0, cells were exposed to 100 μM RV and re-treated (on days 1 and 3 with 20 μM RV; on days 5 and 7, 10 μM RV was added alone or in combination with 50 ng/mL IL-6 or replaced by IL-6) up to 10 days. 3D spheroid dimensions were determined with ImageJ software. Data represent the average ± S.D. calculated for three different fields per condition in three separate experiments. (**A**) Time-course phase-contrast photos of OVCAR3 3D spheroids. (**B**) Growth curves represent the dimensions of 3D spheroids during the experimental timeline. (**C**) Graph reporting the quantification of 3D spheroid areas on day 10. Statistical analysis was performed by using GraphPad Prism 5.0 software. Bonferroni’s multiple comparison test after one-way ANOVA was employed. Significance was considered as follows: * *p* < 0.05.

**Figure 5 cancers-14-02142-f005:**
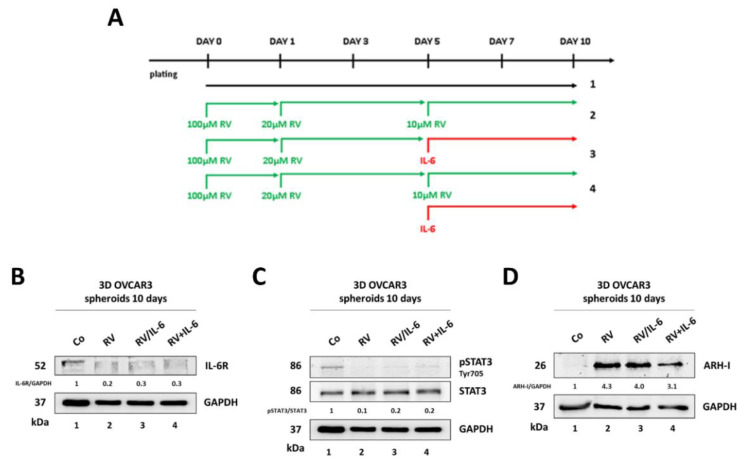
Resveratrol hampers IL-6R/STAT3 signaling in 3D ovarian cancer spheroids. (**A**) Experimental timeline of the 3D spheroids-forming assay. (**B**–**D**) Western blotting showing the expression of IL-6R (**B**), pSTAT3 (Tyr705)/STAT3 (**C**), and ARH-I (**D**) on day 10. The densitometric analysis of Western blotting representative of three replicates is included.

**Figure 6 cancers-14-02142-f006:**
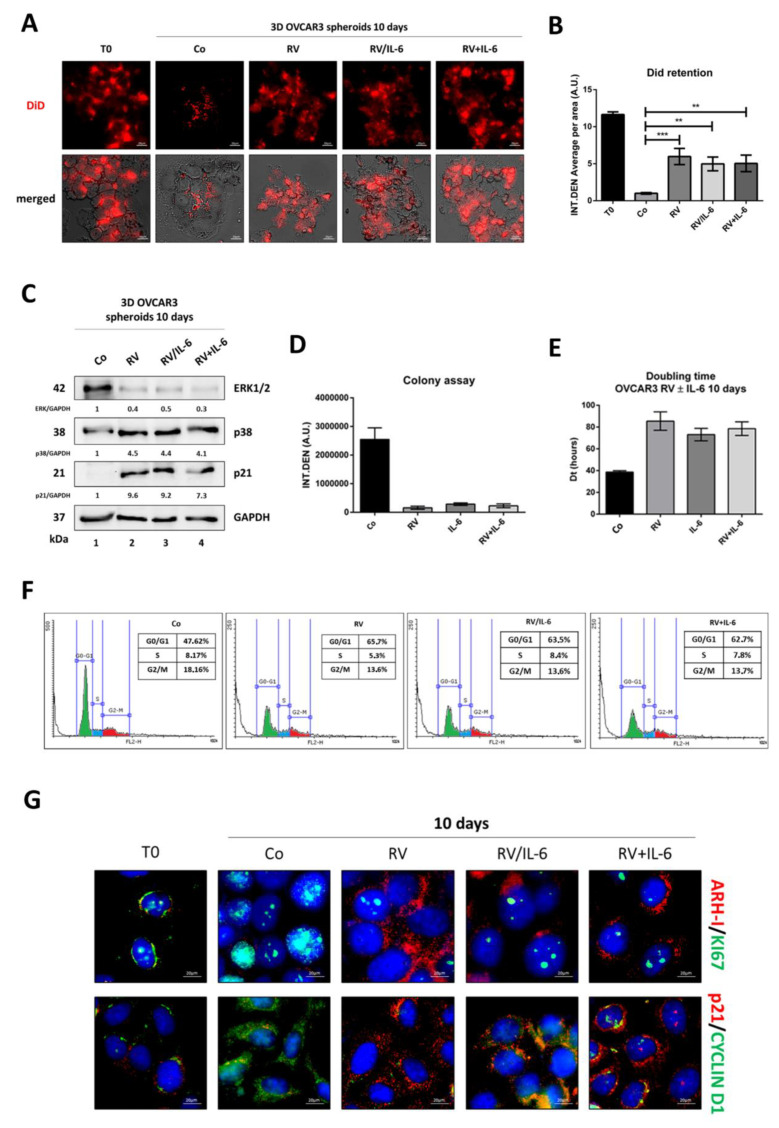
Resveratrol keeps 3D ovarian cancer spheroids in a dormant state. (**A**–**C**) OVCAR3 cells were grown as 3D spheroids and treated following the same timeline as Figure 5. A. Assessment of cell proliferation rate in OVCAR3 3D spheroids by DiD staining. Scale bar = 20 µm; magnification = 63×. (**B**) Quantification of DiD dye retention performed with ImageJ software. Data ± S.D. represent three independent replicates. Significance was considered as follows: *** *p* < 0.001; ** *p* < 0.01. (**C**) Western blotting for the expression of proliferation/quiescence markers (ERK1/2, p38, and p21) in 3D OVCAR3 spheroids collected on day 10. The densitometric analysis of Western blotting representative of three independent experiments is included. (**D**–**G**) OVCAR3 cells were cultured in 2D and treated as above. (**D**) Colony formation assay calculated on day 10. Data ± S.D. are representative of three independent replicates. (**E**) Doubling time calculated on day 10. Data ± S.D. are representative of three independent replicates. (**F**) Cell cycle analysis performed on day 5 and day 10 by using a FacScan flow cytometer. (**G**) Immunofluorescence double-staining performed on day 10 for ARH-I (red)–Ki-67 (green) and p21 (red)–cyclin D1 (green). Scale bar = 20 µm; magnification = 63×.

**Figure 7 cancers-14-02142-f007:**
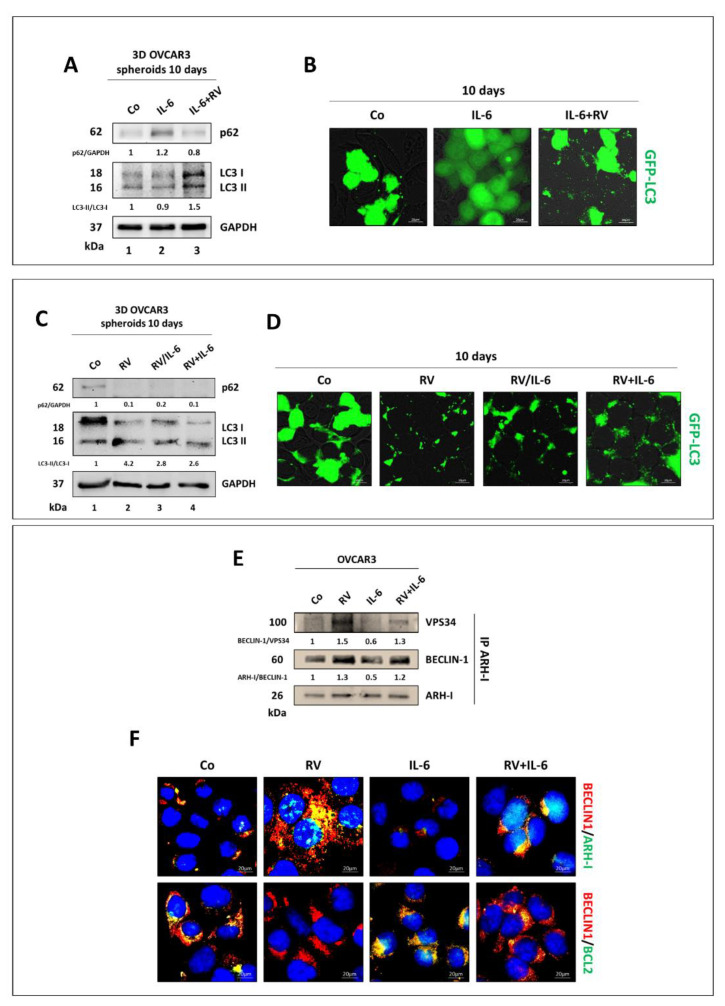
Resveratrol promotes an ARH-I-mediated autophagy. (**A**–**C**) The assessment of autophagy flux modulation in 3D OVCAR3 spheroids cultured for 10 days in the same experimental timeline of Figure 2A (**A**) and Figure 5A (**C**). The Western blotting shows the expression of p62 and the LC3-II/LC3-I ratio. The densitometric analysis of Western blotting representative of three independent experiments is included. (**B**–**D**) OVCAR3-GFP-LC3 cells were treated as per the experimental conditions reported in Figure 2A (**B**) and in Figure 5A (**D**). On day 10, coverslips were imaged with a fluorescence microscope. Scale bar = 20 µm; magnification = 63×. (**E**) Co-immunoprecipitation of ARH-I binding partners belonging to the autophagic initiation complex. OVCAR3 cells were cultured in 2D and treated with 100 μM RV, 50 ng/mL IL-6, or both for 24 h. The quantification of the interactions between the autophagic initiation partners is included. Data are representative of three independent replicates. (**F**) Immunofluorescence double-staining for BECLIN-1 (red)–ARH-I (green) and BECLIN-1 (red)–BCL-2 (green). Scale bar = 20 µm; magnification = 63×.

**Figure 8 cancers-14-02142-f008:**
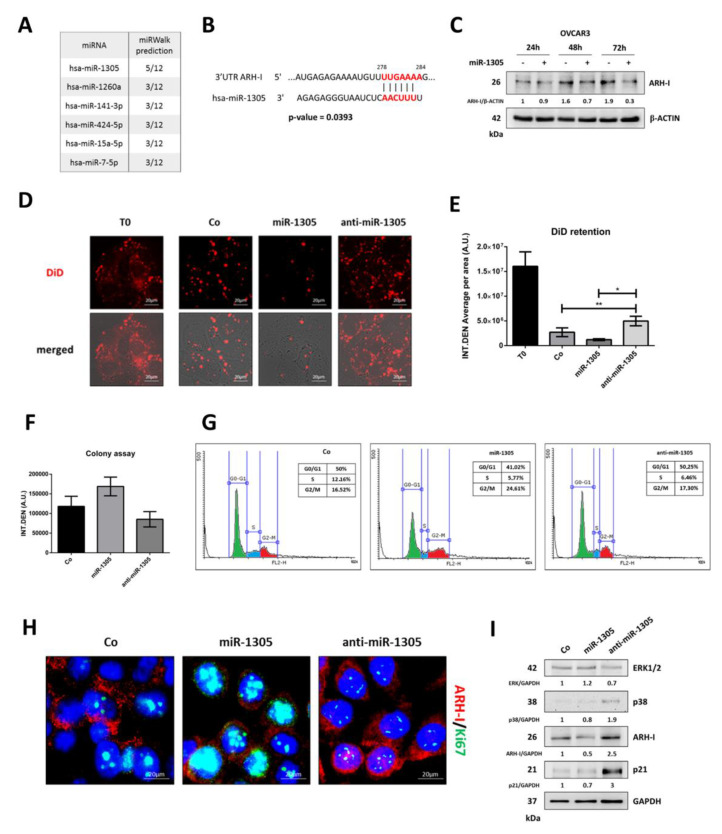
miR-1305 downregulates ARH-I expression, thus sustaining ovarian cancer cell proliferation. (**A**) Table reporting the in silico prediction of miRNA–*ARH-I* interaction performed by the miRWalk 2.0 software on the six miRNAs previously identified to be oppositely modulated by IL-6 and RV. (**B**) Schematic representation of the interaction between hsa-miR-1305 and the 3′UTR of *ARH-I* transcript. The *p*-value of the predicted interaction is included. (**C**) Time-course Western blotting to monitor the effect of miR-1305 mimic on ARH-I protein expression. The densitometric analysis of Western blotting representative of three independent experiments is included. (**D**) Assessment of proliferation rate in OVCAR3 cells transfected with miR-1305 mimic or anti-miR-1305 by DiD dye retention after 72 h. Scale bar = 20 µm; magnification = 63×. (**E**) Graph reporting the quantification of DiD dye retention. Significance was considered as follows: ** *p* < 0.01; * *p* < 0.05. (**F**) Graph representing the colony formation assay performed after 72 h of transfection. Data ± S.D. are representative of three independent replicates. (**G**) Cell cycle analysis performed on day 5 and day 10. Cells were fixed, stained with propidium iodide, and acquired by using a FacScan flow cytometer. (**H**) Immunofluorescence double-staining for ARH-I (red)–Ki-67 (green) in OVCAR3 cells transfected with miR-1305 or anti-miR-1305 for 72 h. Scale bar = 20 µm; magnification = 63×. (**I**) Western blotting analysis for the expression levels of proliferation (ERK1/2) and quiescence markers (p38, ARH-I, p21) in cells transfected with miR-1305 or anti-miR-1305 for 72 h. The densitometric analysis of Western blotting representative of three independent experiments is included.

**Figure 9 cancers-14-02142-f009:**
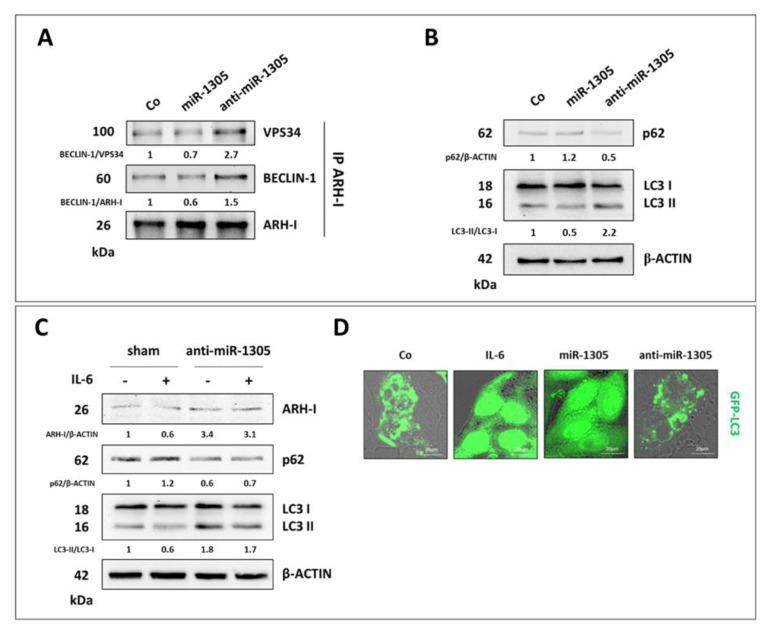
miR-1305 inhibits while anti-miR-1305 induces ARH-I-mediated autophagy. (**A**) The co-immunoprecipitation of ARH-I binding partners of the pro-autophagic interactome. Cells were cultured in 2D and transfected with miR-1305 mimic or anti-miR-1305 for 72 h. The quantification of the protein–protein interactions is included. Data are representative of three independent replicates. (**B**) Western blotting of p62 and LC3 for determining the autophagy flux in OVCAR3 cells transfected with miR-1305 or anti-miR-1305 for 72 h. Densitometric analysis of Western blotting representative of three independent experiments is included. (**C**) OVCAR3 cells were transfected with anti-miR-1305, and after 16 h the cells were treated with 50 ng/mL IL-6 and further cultured for 24 h. Western blotting of p62, ARH-I and LC3. The densitometric analysis of Western blotting representative of three independent experiments is included. (**D**) OVCAR3-GFP-LC3 cells were plated on coverslips, treated with 50 ng/mL IL-6, or transfected with miR-1305 or anti-miR-1305 for 72 h. Coverslips were imaged with a fluorescence microscope. Scale bar = 20 µm; magnification = 63×.

**Figure 10 cancers-14-02142-f010:**
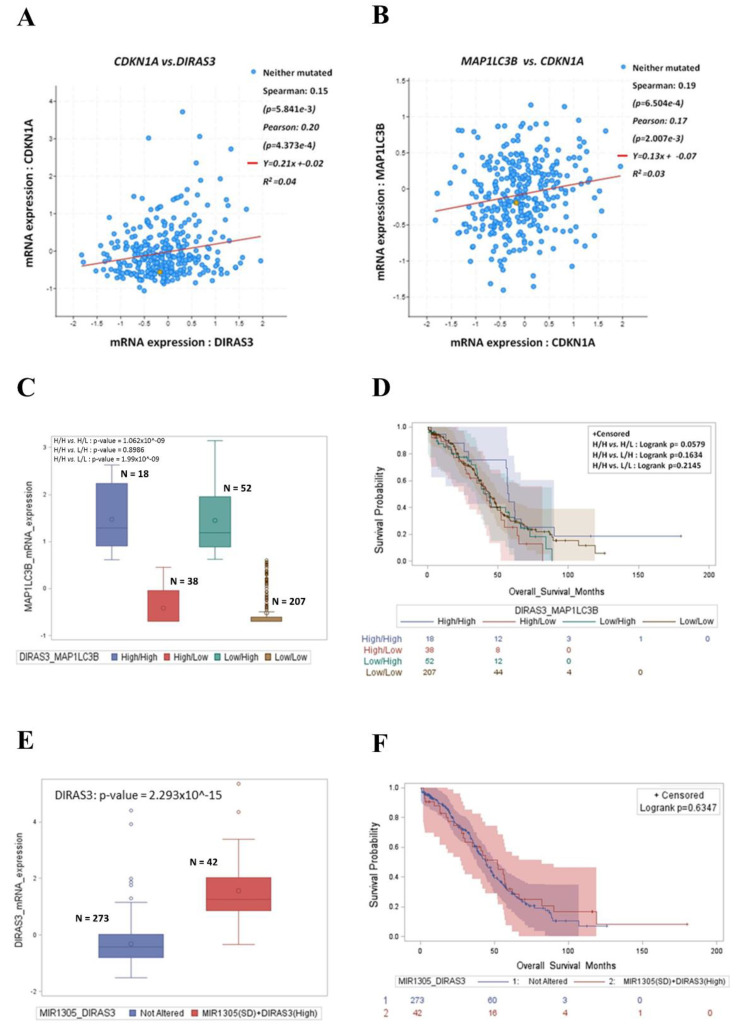
Patients with a high expression of *DIRAS3*/*MAP1LC3B* together with *CDKN1A* upregulation have better overall survival and this correlates with *MIR1305* downregulation. (**A**) Scatter plot representing the correlation between *DIRAS3* and *CDKN1A* mRNA expression. (**B**) Scatter plot representing the correlation between *MAP1LC3B* and *CDKN1A* mRNA expression. The Spearman’s and Pearson’s correlation coefficients and *p*-value were used to determine statistical significance. The *p*-value ≤ 0.05 was considered significant. (**C**) Box-plot showing the distribution of *DIRAS3* mRNA expression according to *MAP1LC3B* mRNA levels (High/High, High/Low, Low/High, Low/Low). (**D**) Kaplan–meier plot representing the overall survival status of patients stratified in the four-group analysis (High *DIRAS3*/High *MAP1LC3B* (H/H), High *DIRAS3*/Low *MAP1LC3B* (H/L), Low *DIRAS3*/High *MAP1LC3B* (L/H), Low *DIRAS3*/Low *MAP1LC3B* (L/L)). The *p*-value ≤ 0.05 was considered significant. (**E**) Box-plot showing the distribution of *DIRAS3* mRNA expression according to *MIR1305* CNV *status* (shallow deletion (SD) *vs*. not altered (includes rest of patients without the shallow deletion of *MIR1305*)). (**F**) Kaplan–meier plot representing the overall survival status of patients stratified in not altered *vs*. SD *MIR1305* + High *DIRAS3*.

## Data Availability

Data sharing is not applicable to this article.

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
