# Peer review of "Resveratrol Contrasts IL-6 Pro-Growth Effects and Promotes Autophagy-Mediated Cancer Cell Dormancy in 3D Ovarian Cancer: Role of miR-1305 and of Its Target ARH-I"

_cancers, 2022, doi:10.3390/cancers14092142_

Round 1
Reviewer 1 Report
The manuscript by Esposito A et al. showed that resveratrol induced autophagy-mediated cancer cell dormancy in ovarian cancer cell lines. The authors revealed that resveratrol treatment canceled IL-6-induced cell growth. Resveratrol and IL-6 oppositely regulated miR-1305, which targeted ARH-1. ARH-1 is known to regulate autophagy. In the manuscript, with resveratrol and IL-6 combination treatment, the authors clearly demonstrated cancer cell dormancy by resveratrol which induced ARH-mediated autophagy. Generally, the necessary experiments were performed, and the manuscript was well-written.
- In Figure 7, both LC3I and LC3II were upregulated in IL-6+RV treatment (Figure 7A); however, in Figure 7C, this reviewer could not see up-regulation of LC3, although the authors mentioned: “LC3-II increased over LC3-I in tumor spheroids exposed to RV”. Please explain this.
Author Response
We thank the reviewers for their positive assessment of our manuscript and for suggesting insightful improvements.
We have taken into considerations all the criticisms and suggestions amending the text accordingly (all alterations in the text are reported in red), as detailed in the point-by-point answers to reviewers’ comments reported below.
Further, while under review we went on with our work and here we provide additional results, which corroborate and further support our findings on the role of miR-1305 in autophagy downregulation (new Figure 9) and on the prognostic value of DIRAS3-mediated autophagy in ovarian cancer patients (Figure 10C-D).
Finally, we took this opportunity to also fix a few grammar/syntax errors.
REVIEWER 1
The manuscript by Esposito A et al. showed that resveratrol induced autophagy-mediated cancer cell dormancy in ovarian cancer cell lines. The authors revealed that resveratrol treatment canceled IL-6-induced cell growth. Resveratrol and IL-6 oppositely regulated miR-1305, which targeted ARH-1. ARH-1 is known to regulate autophagy. In the manuscript, with resveratrol and IL-6 combination treatment, the authors clearly demonstrated cancer cell dormancy by resveratrol which induced ARH-mediated autophagy. Generally, the necessary experiments were performed, and the manuscript was well-written.
- In Figure 7, both LC3I and LC3II were upregulated in IL-6+RV treatment (Figure 7A); however, in Figure 7C, this reviewer could not see up-regulation of LC3, although the authors mentioned: “LC3-II increased over LC3-I in tumor spheroids exposed to RV”. Please explain this.
We thank the reviewer for pointing to this apparent discrepancy.
The increased LC3-II/LC3-I ratio and p62 degradation are consistent with autophagy induction both in panel A and C. The slight difference in quantitative terms is likely due to the different protocol used: in the first case (panel A) RV was added after 5 days of IL-6 treatment, while in the second case (panel C) RV was chronically administered from day 1. It is conceivable that in the latter case the autophagy flux was very much increased by the chronic treatment with RV. To improve readability of this data, we have substituted the blots of Figure 7C with higher exposed ones where LC3 bands are better appreciated.

Reviewer 2 Report
The manuscript is well written. It demonstrated that resveratrol induced autophagy-mediated cancer cell dormancy by applying to three different ovarian cancer cell lines, thus can be presented as a novel therapeutic tool.
The introduction is clear with a nice graphical abstract. The graphic is informative and aesthetic.
However, the methods can be further improved.
There are few comments here:
Page 5, line 157: ImageJ software - version?
Page 5, line 171: VersaDOC Imaging System – version? City? Country?
Page 5, line 173: Quantity One Software – version? City? Country?
Page 5, line 186: FacScan flow cytometer – model? City? Country?
Page 5, line 188: Flowing software 2.0 - city? Country? Downloaded from? The latest version from https://bioscience.fi/services/cell-imaging/flowing-software/ is 2.5.1. Same software?
Page 7, line 242: ….was performed with the software ImageJ -> was performed using ImageJ software.
Page 7, line 245: Any reason why GraphPad Prism 5.0 software was used for the statistical analysis, which is different from page 6, line 234: R and SAS software?
For results section, the statistical analyses were presented in Figure 1, Figure 3, Figure 4, Figure 6, Figure 8, and Figure 9. All kind of results cramped into one figure which could be too crowded. Any possibility to present the statistical results in table form?
This manuscript has scientific merit and provide new information to current knowledge.
Author Response
We thank the reviewers for their positive assessment of our manuscript and for suggesting insightful improvements.
We have taken into considerations all the criticisms and suggestions amending the text accordingly (all alterations in the text are reported in red), as detailed in the point-by-point answers to reviewers’ comments reported below.
Further, while under review we went on with our work and here we provide additional results, which corroborate and further support our findings on the role of miR-1305 in autophagy downregulation (new Figure 9) and on the prognostic value of DIRAS3-mediated autophagy in ovarian cancer patients (Figure 10C-D).
Finally, we took this opportunity to also fix a few grammar/syntax errors.
REVIEWER 2
The manuscript is well written. It demonstrated that resveratrol induced autophagy-mediated cancer cell dormancy by applying to three different ovarian cancer cell lines, thus can be presented as a novel therapeutic tool.
The introduction is clear with a nice graphical abstract. The graphic is informative and aesthetic.
However, the methods can be further improved.
There are few comments here:
Page 5, line 157: ImageJ software - version? Done
Page 5, line 171: VersaDOC Imaging System – version? City? Country? Done
Page 5, line 173: Quantity One Software – version? City? Country? Done
Page 5, line 186: FacScan flow cytometer – model? City? Country? Done
Page 5, line 188: Flowing software 2.0 - city? Country? Downloaded from? Done The latest version from https://bioscience.fi/services/cell-imaging/flowing-software/ is 2.5.1. Same software? Yes
Page 7, line 242: ….was performed with the software ImageJ -> was performed using ImageJ software. Done
Page 7, line 245: Any reason why GraphPad Prism 5.0 software was used for the statistical analysis, which is different from page 6, line 234: R and SAS software? We employed GraphPad Prism 5.0 to draw the graphs and perform the statistical analysis of data belonging to benchwork. R and SAS software were employed while performing the bioinformatic analysis of TCGA data regarding the patients’ survival outcome.
For results section, the statistical analyses were presented in Figure 1, Figure 3, Figure 4, Figure 6, Figure 8, and Figure 9. All kind of results cramped into one figure which could be too crowded. Any possibility to present the statistical results in table form? We prefer to present the statistical significance in the graphical form to save space and to simplify the comparisons for the readers. The table form may be difficult to be seen in the final version of the figures (that usually are scale down during the editorial proofreading).
This manuscript has scientific merit and provide new information to current knowledge. Thanks for your kind words.
